# Evaluation of PD-L1 Expression in Colorectal Carcinomas by Comparing Scoring Methods and Their Significance in Relation to Clinicopathologic Parameters

**DOI:** 10.3390/diagnostics14101007

**Published:** 2024-05-13

**Authors:** Mirela Frančina, Mislav Mikuš, Marin Mamić, Tihomir Jovanović, Mario Ćorić, Božica Lovrić, Ivan Vukoja, Goran Zukanović, Kristijan Matković, Jasmina Rajc, Ferdinand Slišurić, Mateja Jurić-Marelja, Goran Augustin, Ilijan Tomaš

**Affiliations:** 1Požega General Hospital, 34000 Požega, Croatia; mirela.francinamf@gmail.com (M.F.); mmamic@fdmz.hr (M.M.); bozica.lovric@pozeska-bolnica.hr (B.L.); kristijan.matkovic@pozeska-bolnica.hr (K.M.);; 2Faculty of Medicine, University of J.J. Strossmayer, 31000 Osijek, Croatia; tihomir.jovanovic@ozbpakrac-bhv.hr (T.J.); ilijan.tomas@mefos.hr (I.T.); 3Department of Obstetrics and Gynecology, University Hospital Centre Zagreb, 10000 Zagreb, Croatia; mcoric77@gmail.com; 4General Hospital Pakrac, 34550 Pakrac, Croatia; 5Faculty of Dental Medicine and Health, University of J.J. Strossmayer, 31000 Osijek, Croatia; 6Clinical Hospital Center Osijek, 31000 Osijek, Croatia; 7Department of Surgery, University Hospital Centre Zagreb, 10000 Zagreb, Croatia; augustin.goran@gmail.com

**Keywords:** colorectal carcinoma, PD-L1, tumor proportion score, combined positive score

## Abstract

Background: This study aims to evaluate PD-L1 expression in colorectal carcinomas (CRCs) by using the tumor proportion score (TPS) and the combined positive score (CPS), and to investigate whether there is a correlation with clinicopathologic features. Methods: A cross-sectional study was conducted that included samples from patients with colorectal adenocarcinoma treated with colon resection and rectal resection after neoadjuvant radio- and chemotherapy at the Department of Abdominal Surgery at Požega Hospital in the period from 2017 to 2022. The study included 102 tumor tissue samples from patients after resection and the pathohistological diagnosis of adenocarcinoma. Results: In our study, the PD-L1 positivity rate after the TPS was 42 (41%) samples, and after the CPS, 97 (95%) of them (*p* < 0.001). The positive expression of PD-L1 in tumor cells using the TPS method showed a statistically significant association with adenocarcinoma (TPS ≥ 10–50% and ≥50%). There were significantly more that were moderately differentiated, with TPS ≥ 50%, and those poorly differentiated had values ≥ 10–50%. There were significantly more patients with a status of more than one positive lymph node with TPS values ≥ 10–50%. Patients without metastases in the lymph nodes are significantly more likely to have CPS values > 50%, compared with other lymph node statuses. Conclusions: These results suggest that the total number of PD-L1-expressing cells, including tumor and immune cells, is a more sensitive biomarker than the number of PD-L1-expressing tumor cells alone in CRC.

## 1. Introduction

Colorectal cancer (CRC) is a serious health problem and one of the most common diseases leading to morbidity and mortality in all parts of the world [1]. Chemotherapy is one of the most important treatment options for patients with CRC, but its use leads to severe side effects and resistance [1]. In recent years, anti-PD-L1 immunotherapy has been developed as a fourth line of treatment for CRC patients [2].

The immunohistochemical expression of PD-L1 is used as a prognostic biomarker with which to screen patients and decide on immunotherapy for different tumor types [2]. Due to the heterogeneity, genetic, and epigenetic characteristics of CRC, the selection of patients for immunotherapy is based on the assessment of the MMR or MSI status and evidence of metastatic distant disease [3]. Due to the complex etiopathogenesis of CRC, which is associated with genetic and epigenetic alterations, most CRCs remain microsatellite-stable, while only 12–15% are microsatellite-unstable (dMMR) [4]. Metastatic disease in CRC significantly reduces overall survival and survival without disease recurrence [1].

The expression of PD-L1 has been proposed as a prognostic biomarker in colorectal cancer; still, it has not yet been put into practice because the evaluation of immunohistochemical analysis, the scoring method, and the application of different types of immunohistochemical tests, materials, and patient screening are not standardized [5].

Recently, two scoring systems have been developed to evaluate PD-L1 expression: the tumor proportion score (TPS) and the combined positive score (CPS) [6,7]. In the TPS, PD-L1 expression in tumors is assessed based on the ratio of PD-L1-positive tumor cells to the total number of viable tumor cells [8]. The expression percentage of the TPS and CPS was developed to assess programmed death ligand 1 (PD-L1) expression and has been approved by the FDA to aid in identifying patients for immunotherapy treatment for six tumor indications at clinically confirmed CPS diagnostic limits: adenocarcinoma of the stomach or gastroesophageal junction (GC/GEJ) (CPS ≥ 1), cervical cancer (CPS ≥ 1), urothelial carcinoma (CPS ≥ 10), squamous cell carcinoma of the head and neck (HNSCC) (CPS ≥ 1), esophageal squamous cell carcinoma (ESCC) (CPS ≥ 10), and triple-negative breast cancer (TNBC) (CPS ≥ 10) [9,10].

These two scoring systems have not yet been used in clinical studies on the therapeutic effect of PD-1 inhibitors on CRC, and there is no standardized scoring system for them. Previously published studies evaluating the immunohistochemical expression of PD-L1 in CRC have reported conflicting results regarding the percentage of expression, impact on survival, and correlation with clinicopathologic features; therefore, this study aims to evaluate PD-L1 expression by using these two scoring systems and to investigate whether there is a correlation with clinicopathologic features. 

## 2. Materials and Methods

### 2.1. Study Characteristics

A cross-sectional study was conducted that included samples from patients with colorectal adenocarcinoma treated with colon resection and rectal resection after neoadjuvant radio- and chemotherapy at the Department of Abdominal Surgery at Požega Hospital. The study included 102 tumor tissue samples from patients after resection and the pathohistological diagnosis of adenocarcinoma at the Department of Pathology and Cytology of Požega Hospital from 2017 to 2022. 

All patients with colorectal adenocarcinoma who were diagnosed and treated with colon resection and rectal resection after neoadjuvant radio- and chemotherapy were included in the study, regardless of subtype, stage, gender, and age. Malignant tumors of non-epithelial origin were excluded from the sample (e.g., neuroendocrine tumors, lymphomas, and gastrointestinal stromal tumors). Immunohistochemical staining for DNA repair proteins was performed on all samples, and additional molecular analyses of KRAS, NRAS, and BRAF genes were performed on some samples from paraffin block samples by using the polymerase chain reaction (PCR) at the Clinical Institute of Pathology of KBC Osijek. To have enough tissue for the immunohistochemical analysis, resection samples were used, while diagnostic small biopsy samples were excluded.

Data were collected on the histological type, grade, tumor size, growth method, presence of tumor budding, TNM stage, presence of intratumoral lymphocytes (TILs), presence of tumor-associated macrophages (TAMs), presence of lymphocapillary and perineural invasion, presence of focal clusters of lymphocytes with germinal centers on the periphery (Chron’s reaction type), lymph node status, presence of distant metastases, MMR status, and existence of BRAF and RAS mutations. Patients were followed until death or the end date of the study (May 2022). Ethical approval was obtained from the Ethics Commission of the General Hospital Požega and the Ethics Commission of the Faculty of Medicine in Osijek.

### 2.2. Evaluation

The material was processed according to a standard procedure for obtaining pathohistological samples, which includes the fixation of the tissue in 10% buffered formalin between 12 and 72 h, sectioning to a thickness of 5 µm, deparaffinization, and staining with the standard hemalone–eosin (HE) method. Morphologically distinct areas of each tumor and various morphological parameters were identified, including the tumor location, tumor size, growth pattern (infiltrating or pushing), degree of invasion, histological type of tumor, grade, the number of TILs, focal clusters of lymphocytes with germinal centers at the periphery (Chron’s reaction type), lymphatic invasion, perineural invasion, the presence of necrosis, the presence of tumor cell deposits in the pericolic adipose tissue, and the presence of tumor buds. For the immunohistochemical analysis of PD-L1 expression and the immunohistochemical analysis of DNA repair proteins from the archives of the Department of Pathology and Cytology of the Požega General Hospital, a cube of adenocarcinoma and intestinal wall tissue not affected by the tumor was separated from each patient and the properties of the mentioned markers in the whole preparation were analyzed. From the paraffin blocks, after selecting a representative region of the tumor on the histological sections stained with H&E, a section was isolated from the tumor together with the tumor stroma without areas of necrosis and the surrounding normal mucosa for immunohistochemical analysis. An area of deepest invasion, an area at the border between the tumor and tumor stroma, and an area on the surface of the mucosa affected by the tumor were analyzed. The slides stained with hematoxylin and eosin were examined by two pathologists. 

Two pathologists assessed the TIL status by using matched slides stained with H&E. One area from the site of the deepest invasion, one area at the junction of the tumor and the tumor stroma, and one area on the surface of the mucosa affected by the tumor containing infiltrating immune cells were analyzed. The number of TILs was determined based on the recommendation for determining the features that indicate microsatellite instability: none, few to moderate (0–2 for one large field of view of 400× magnification), and many (3 or more for one large 400× magnification field of view). The results were semiquantitatively assessed on a four-point scale: result 1 (no infiltrating lymphocytes), result 2 (small or moderate increase in infiltrating lymphocytes), and result 3 (pronounced increase in infiltrating lymphocytes). We determined the number of TAMs on the following basis: low (0–10 for one large field of view at 400× magnification) and many (more than 10 for one large field of view at 400× magnification). The results were evaluated on a two-point scale: result 1 (small increase in tumor-related macrophages) and result 2 (pronounced increase in tumor-related macrophages). 

An immunohistochemical analysis of DNA repair proteins was performed on the histological samples of paraffin blocks from patients who underwent a surgical resection of colorectal cancer at Požega General Hospital. Representative blocks of tissue fixed in paraffin were then selected, showing part of the tumor tissue and part of the adjacent normal intestinal mucosa unaffected by the tumor. A panel with four MMR protein antibodies (MLH1, MSH2, MSH6, and PMS2) was performed using the DAKO En Vision method on representative tissue blocks fixed in paraffin. Immunohistochemistry was performed on a DAKO Autostainer Link 48 automatic counter. According to the CAP protocol for immunohistochemistry interpretation, any nuclear staining, even when spotty, is considered a “no loss of expression” finding. Only an absolute lack of nuclear staining should be considered a “loss of expression”, provided the internal controls are positive. A positive internal control is considered positive nuclear staining of the epithelial cells of the normal colonic mucosa. The expression of the proteins described above was grouped into the following four categories: no loss of expression, loss of expression of all four proteins, combined loss of expression of two proteins, and isolated loss of expression of only one protein. The immunohistochemical analysis of PD-L1 was performed using clone SP263 Ventana on 5 µm tissue sections from paraffin blocks. All immunohistochemically stained sections were independently analyzed by two authors for the tumor proportion score (TPS) and combined positive score (CPS). 

### 2.3. Tumor Proportion Score PD-L1 Expression Determination and Estimation by the TPS and CPS

The immunohistochemical analysis of PD-L1 was performed using clone SP263 Ventana (equivalent assay for clone 22C3) on 5 µm tissue sections from paraffin blocks.

A PD-L1 (SP 263) assay immunohistochemical test using a rabbit monoclonal anti-PD-L1 primary antibody (VENTANA PD-L1 (SP263) antibody) was used to recognize programmed cell death ligand 1 (PD-L1), also known as B7 homolog (B7-H1) or CD 274.

Immunohistochemistry was performed on a VENTANA BenchMark XT using a kit according to the manufacturer’s instructions with built-in deparaffinization, retrieval, and staining [11]. Two authors independently analyzed all immunohistochemically stained sections to evaluate the percentage of positive tumor cell expression (TPS) and the percentage of positive tumor and immune cell expression (CPS). Two pathologists also independently performed a microscopic interpretation of PD-L1 expression.

The criterion for positive immunohistochemical staining was as follows: Cells with positive cytoplasmic and membrane expression were manually counted by semiquantitative evaluation with a light microscope. In contrast, any absence of any visible PD-L1 staining or visible PD-L1 staining membrane as well as cytoplasmic expression of any intensity and that occupy < 1% of the tumor surface are considered negative.

The degree of staining intensity (from 1+ to 3+) was included in the scoring.

PD-L1 is considered immunohistochemically positive when membranous or cytoplasmic staining of any intensity equal to or more than 1% of the tumor surface (tumor cells, immune cells in intratumor and adjacent peritumor stroma) is visible, with a PD-L1 expression level ≥1%.

The specified threshold values of ≥1% are based on previously published studies [12,13,14,15]. PD-L1 expression was evaluated in the tumor and associated stroma, and the tumor did not affect the colon mucosa.

In the H&E preparation taken from the tumor area, we assessed the adequacy of the sample (tumor cells, immune cells, and intratumoral as well as adjacent peritumoral stroma). We assessed whether there were areas of necrosis.

We included a minimum of 100 viable tumor cells with tumor stroma.

The assessment of PD-L1 expression using TPS and CPS on the entire preparation was analyzed first at small magnifications (4×)—an assessment of the expression of tumor and immune cells. Then, at higher magnifications (20×), we determined the total number of PD-L1-positive and -negative viable tumor cells as well as the number of PD-L1-positive tumor and immune cells.

#### 2.3.1. How the TPS Is Determined

The TPS (%) is the number of PD-L1-positive tumor cells/total number of viable tumor cells × 100. This scoring method estimates the percentage of viable tumor cells that show partial or complete staining of the membrane of any intensity against the total number of viable tumor cells. The TPS determines the level of PD-L1 expression, which is reported as a percentage on a scale from 0% to 100%. A minimum of 100 viable tumor cells per PD-L1-stained slide are required for a sample to be considered suitable for PD-L1 assessment [7,10].

#### 2.3.2. How the CPS Is Determined

The CPS is the number of PD-L1-positive cells (tumor and immune)/total number of viable tumor cells × 100. Although the CPS calculation score can exceed 100, the maximum score is defined as a CPS of 100 [7,10]. A sample must contain at least 100 viable tumor cells per PD-L1-stained slide to be considered suitable for PD-L1 assessment [10].

We used five cut-off values of the TPS and CPS currently used for the immunohistochemical assessment of PD-L1 in both tumor and immune cells, and classified them as follows:

The TPS is scored with numbers from 1 to 5 (1 < 1%, 2 ≥ 1–5%, 3 ≥ 5–10%, 4 ≥ 10–50%, and 5 > 50%). The CPS is scored with numbers from 1 to 5 (1 < 1, 2 ≥ 1–5, 3 ≥ 5–10, 4 ≥ 10–50, and 5 > 50).

We excluded from the numerators uncounted tumor cells, tumor cells with only a cytoplasmic reaction, other benign epithelial cells, necrotic cells and cellular detritus, unstained immune cells, immune cells associated with benign structures, lymphoid aggregates that are not directly involved in the tumor response, neutrophils, eosinophils, and plasma cells.

When determining the examined samples, we divided the heterogeneous tumor area into parts with an equal number of tumor cells. We have determined the CPS and TPS for each area.

CPS = PD-L1-positive cells (tumor, lymphocytes, and macrophages)/the total number of viable tumor cells × 100 [10].

TPS = PD-L1-positive tumor cells/the total number of PD-L1-positive and PD-L1-negative tumor cells × 100 [7,10].

### 2.4. Sample Characteristics

This study included 102 cases of CRC: 76 (74.5%) samples were from male patients, and 26 (25.5%) were from female patients. With regard to age, 79 (77.5%) samples were from patients aged 65 years, and 23 (22.5%) samples were from patients up to 65 years of age. Regarding the localization, slightly more tumors, 44 (43.1%), were located on the left side; 36 (35.3%) were on the right side and 22 (21.6%) were located in the rectum. Most of the tumors, 90 (88.2%), were adenocarcinomas (NOS tip), while 11 (10.8%) were mucinous and 1 was a medullary type of colon cancer. According to the degree of differentiation, 83 (81.4%) were moderately differentiated. As for the size of the tumor, a tumor of size 3 to 5 cm is found in 54 (52.9%) samples, larger than 5 cm in 36 (35.3%), and smaller than 3 cm (10.8%) in 11 samples. The infiltrating growth mode is present in 78 (76.5%) samples and tumor budding in 13 (12.7%) samples. According to the TNM classification, most samples are grade II 28 (27.5%) and III 57 (55.9%). Tumor-infiltrating lymphocytes (TILs) are low or moderate in 72 (70.6%) samples, and high in 28 (27.5%). Few tumor-associated macrophages (TAMs) are found in 89 (87.3%) samples, and many macrophages are found in 13 (12.8%) samples. Many focal accumulations of lymphocytes with germinal centers on the periphery are observed in 45 (44.1%) samples. Lymphovascular and perineural invasion was present in 33 (32.3%) samples. Negative lymph nodes were found in 67 (65.7%) samples. Sixteen (15.7%) patients had metastases in only one organ (liver), and only one patient (1%) had metastases in two or more organs.

### 2.5. Statistical Analysis

Categorical data are represented by absolute and relative frequencies. Differences in categorical variables between independent groups were tested with the χ^2^ test, and, if necessary, Fisher’s exact test. Differences in dependent categorical variables were tested with the McNemar–Bowker or marginal homogeneity test. The normality of the distribution of continuous variables was tested with the Shapiro–Wilk test. All *p*-values are two-sided. The significance level was set at *p* = 0.05. For statistical analyses, the statistical package MedCalc^®^ Statistical Software version 22.016 (MedCalc Software Ltd., Ostend, Belgium; https://www.medcalc.org; (accessed on 25 November 2023) 2023) and SPSS 23 (IBM Corp. Released 2015. IBM SPSS Statistics for Windows, Version 23.0. IBM Corp., Armonk, NY, USA) were used.

## 3. Results

### 3.1. Assessment of PD-L1 Expression Using the TPS and CPS

PD-L1 expression was assessed in each sample using the TPS and CPS methods. In 60 (58.8%) cases, it was negative (<1%) using the TPS method, compared to 5 (4.9%) using the CPS method. Using the TPS, only two (2%) samples had positivity higher than 50%. Using the CPS method, samples with 5 to 10% positive cells had the highest percentage of positivity 36 (35.3%). 

Comparing the TPS with the CPS, it is observed that there is a significant difference in the distribution of patients according to the positive findings concerning the groups so that 42 (41%) patients were positive according to the TPS, and 97 (95%) of them according to the CPS, which is a significant difference (*p* < 0.001). Comparing the two methods (the TPS vs. the CPS), we cannot say that there is agreement in the results (κ = 0.07 with 95% CI from 0.008 to 0.13) (Table 1) (Figure 1 and Figure 2).

### 3.2. Correlation of PD-L1 Positivity according to Each Scoring Method with Clinical–Pathological Parameters

The distribution of features according to the TPS and CPS are demonstrated in Table 2 and Table 3. There are significant differences in the TPS and CPS of the entire sample in the group of men (*p* < 0.001) and the group of women (*p* < 0.001). In both groups, in relation to age, there are significantly lower percentages of the TPS at the site of the deepest invasion (*p* < 0.001), and the percentage of the CPS in the area of the deepest invasion is significantly higher (*p* < 0.001). Concerning localization, both in right-sided (*p* < 0.001) and left-sided tumors (*p* < 0.001), as well as in rectal tumors (*p* < 0.001), there were significantly lower percentages of the TPS at the site of deepest invasion. The representation of CPS at the site of the deepest invasion was significantly higher (Figure 3).

### 3.3. Correlation of Positivity of Both Scoring Systems with MMR Status

Regarding MMR status, the lack of one protein is recorded in 17 (16.7%) samples and two or more proteins in 19 (18.6%) samples. Out of a total of 60 (59%) patients with a TPS < 1%, there are significantly fewer, 8 (42%), with a deficiency of two or more proteins. In comparison, there are significantly more patients with a deficiency of two or more proteins with a TPS of 1–5%. TPS values of 10–50% are significantly higher in patients with one protein deficiency (*p* = 0.02). There is no significant association between the CPS and MMR status.

### 3.4. Correlation of Positivity of Both Scoring Systems with RAS and BRAF Mutations

An RAS mutation was present in two (2%) patients and a BRAF one in four (3.9%). There is no significant difference in the distribution of patients according to the percentage of the TPS and the CPS of the entire sample in patients who do not have an RAS or BRAF mutation. Patients with an RAS (*p* < 0.001) or BRAF (*p* < 0.001) mutation in the TPS have significantly lower percentages compared to the CPS, whose percentage is significantly higher. There is no significant difference in the distribution of patients according to the percentage of the TPS at the site of the deepest invasion or the CPS at the site of the deepest invasion in patients who do not have an RAS or BRAF mutation. In patients with an RAS (*p* < 0.001) or BRAF (*p* < 0.001) mutation, the TPS has significantly lower percentages compared to the CPS, whose percentage is significantly higher. There is no significant difference in the distribution of patients according to the percentage of the TPS at the tumor/tumor stroma transition or the CPS at the tumor/tumor stroma transition in patients who do not have an RAS or BRAF mutation. In patients with an RAS (*p* < 0.001) or BRAF (*p* < 0.001) mutation at the tumor/tumor stroma transition point, significantly lower percentages of the TPS are present compared to the CPS, whose percentage is significantly larger.

## 4. Discussion

One of the main findings of our study is that a statistically significant proportion of respondents were negative for PD-L1 at the TPS assessment (TPS < 1%), while only 5% of respondents were below 1% at the CPS assessment. The highest number was 5–10%, which is statistically significant compared to other percentages. There were significantly more patients with lymph node status 2 (more than four positive lymph nodes) with TPSs ≥ 10–50% (*p* = 0.005) compared with other lymph node statuses. This means that lower PD-L1 values are associated with some worse prognostic parameters, such as the CPS in positive lymph nodes (higher score in negative nodes), as well as less differentiated cancers with lower TPS values and a higher percentage of negative scores in microsatellite-stable tumors, which are known to have a worse prognosis.

In CRC, according to current knowledge on the evaluation of candidates for immunotherapy treatment, it was determined that the highest response rate is achieved when CRC simultaneously shows microsatellite instability, a high burden of tumor mutations, and abundant accumulations of lymphocytes infiltrating the tumor [16]; however, only a small number of CRCs belong to that category, and there is a need to find another prognostic biomarker that would be applicable in daily clinical practice, simple, accessible to most pathohistological laboratories in the world, and not very expensive [17].

Recently, two scoring systems have been developed to evaluate PD-L1 expression: the tumor proportion score (TPS) and the combined positive score (CPS) [6,7]. Controversial results in the studies published so far arise from the lack of a uniform scoring system, positivity thresholds, different clones, and the different screening of samples and patients [6,7]. In this study, we compared the two scoring systems, the TPS and the CPS, for assessing immunohistochemical PD-L1 positivity and correlated them with clinicopathologic features, MMR status, and RAS as well as BRAF mutations. In our study, the positivity rate after the TPS was 42 (41%) samples and after the CPS 97 (95%), which is a significant difference (*p* < 0.001).

The results show that the assessment of PD-L1 by the CPS method is a more sensitive biomarker than the assessment by the TPS method in CRC [18,19]. These results suggest that the expression assessment in tumor cells alone may not reflect the overall immune tolerance in the tumor [18]. With many tumor-infiltrating lymphocytes (TILs), the CPS values are more significant at >50% compared to little or no tumor infiltration, which is in agreement with previously published studies [20,21]. These results once again support the results from previous studies and the conclusions that lymphocytes infiltrating the tumor, regardless of the type of T- or B-cells, play a significant role in the immune microenvironment, and further research on the resolution of their role could be the basis for the realization of personalized immunotherapy for patients with CRC [22].

Other studies reported significantly lower positivity rates, but this study considered both the staining intensity and the percentage of positive tumor and immune cells in the areas with the highest intensity [11,12,23]. This study also used Ventana clone 263, which has the highest sensitivity compared to other clones, according to previously published studies [11,12,23]. In addition, membrane and cytoplasmic staining in immune cells (lymphocytes and macrophages) were considered in this study. Our method complies with the manufacturer’s recommendations and is consistent with previous studies. In this work, the expression of PD-L1 is higher in men regardless of the scoring method, but this is only statistically significant when scoring is carried out with the CPS method. Additionally, patients younger than 65 years in our study have higher expression with both methods, but this was only statistically significant with the TPS method in terms of the percentage of tumor cell expression ≥ 10–50%. In short, the positive expression of PD-L1 in tumor cells using the TPS method showed a statistically significant association with adenocarcinoma (TPS ≥ 10–50% and ≥50%). 

Our study found greater positivity with both methods in tumors of higher stages (II, III, and IV), although this was not statistically significant. The TPS method showed a better correlation with clinicopathological features; it is significantly higher in adenocarcinomas, poorly differentiated tumors, and metastases in more than one lymph node. Both methods showed significantly higher expression in men and younger patients. Sixty-six (64.7%) of our samples were tumors without MMR defects. Two or more proteins are defective in 19 (18.6%) samples, while one is defective in 17 (16.7%). This is a higher percentage compared to the reported results of 15% [21]; however, studies from Egypt (36 and 37%) and Jordan (22%) report higher percentages [21,22]. This study did not find a significant association between CPSs and MMR status. This partially contradicts previously published studies reporting a higher expression of PD-L1 by the CPS method in tumors with MSI/d MMR; however, others report results similar to our study [22]. Unfortunately, it is difficult to determine which of our samples might have Lynch syndrome. Still, the samples with a loss of MMR protein indicate a higher percentage of Lynch syndrome than expected and certainly much higher than that reported in the literature.

These results are in accordance with the results of other studies that state that tumors with increased PD-L1 expression mostly show a lack of MMR protein and are associated with worse clinical–pathological parameters and, thus, a worse prognosis.

MMR-protein-deficient tumors are predisposed to the accumulation of mutations with an increased probability of neoantigen expression and the appearance of strong immunogenicity.

Our results show that the percentage of positive tumor cells (TPS) in CRC is associated with somatic mutations—hypermethylation characteristic of sporadic CRC.

Several reasons may explain why dMMR carcinomas show a higher immunohistochemical expression of PD-L1. The first reason is that the increased number of mutation-associated neoantigens in dMMR tumors can stimulate antitumor immune responses, which can be further enhanced via immune checkpoint inhibition by PD-1 blockading. Another reason is that a higher mutational burden in MSI-H tumors correlates with a higher prevalence of tumor-infiltrating lymphocytes (TILs) that could contribute to an increased antitumor cytotoxic immune response [6,13]. The third reason is that dMMR endometrial carcinomas have been shown to have a significantly increased expression of PD-L1 in tumors and immune stromal cells compared to pMMR carcinomas [6,13]. The fourth reason is that different signaling pathways between dMMR and pMMR tumors can lead to differences in the secretion of factors that activate the PD-1 pathway within the tumor microenvironment. Changes in the TGF-β signaling pathway may be another reason for responses to anti-PD-1 therapy. TGF-β signaling plays a role in immune modulation, and the gene encoding the type II TGF-β receptor is often mutated in dMMR colon cancers [6].

In this study, we found no significant statistical difference in the expression of PD-L1 by the TPS and CPS methods with respect to RAS and BRAF mutations. Patients with an RAS or BRAF mutation have significantly lower percentages with the TPS method than the CPS method, whose percentage is significantly higher [11]. Previous studies have reported a higher expression of PD-L1 in tumor cells in the presence of a BRAF mutation, and further investigation is needed [14].

KRAS and BRAF mutations are generally mutually exclusive in colorectal tumors. Recent studies suggest that BRAF mutations can also be used as predictive markers for EGFR-targeted therapy. Our results are based on a small number of samples included in the study. We associate the differences in these two scoring methods with the fact that very significant immune processes occur in tumors that are burdened with gene mutations and that they, therefore, contain a greater number of immune cells due to the host’s adaptive immune response.

Our work has limitations because it is oriented only to comparing the immunohistochemical expression of PD-L1 in relation to clinical–pathological parameters in CRC without insight into the disease’s outcome, given that our patients did not receive anti-PD-L1 therapy.

Based on the obtained results in our study, most samples showed a percentage of positivity between 5 and 10: 36 of them (35.3%); therefore, we believe that our proposal for the threshold value for assessing PD-L1 positivity in CRC would be values ≥ 10, although the cut-off value for gastrointestinal tract cancers approved by the FDA using the CPS method is ≥1. We believe that such patients could benefit most from immunotherapy treatment [12].

This is probably because, in our cohort, most tumors are in stages II and III and are moderately differentiated tumors. As far as localization is concerned, more are localized on the left side. Compared to previous studies that showed equal or similar representation, this would mean that the total number of patients with these characteristics dominates in relation to higher-stage, poorly differentiated tumors on the right side, where the percentage of PD-L1 expression is significantly higher.

When comparing the two methods (the TPS vs. the CPS), we cannot claim agreement in the results (Cohen’s Kappa κ = 0.07 with a 95% confidence interval of 0.008 up to 0.13).

The result of the Kappa coefficient can be interpreted as follows: values ≤ 0 indicate no agreement, while those in the range of 0.01 to 0.20 indicate no to a slight agreement, 0.21–0.40 indicate fair agreement, 0.41–0.60 represents moderate agreement, 0.61–0.80 indicates significant agreement, and 0.81–1.00 suggests almost perfect agreement. In this case, a Kappa value of 0.07 suggests that the agreement between the methods is small. Still, if other factors are considered, we conclude that the CPS method is a more sensitive biomarker than the TPS method.

In conclusion, in this study, we determined that the immunohistochemical assessment of PD-L1 in tumor cells according to the TPS method of CRC was correlated with the location of tumors on the right side, the status of high microsatellite instability (MSI-H), weaker differentiation, higher pathological T stage, the existence of distant metastases, a higher TNM stage, and the existence of lymphatic as well as perineural invasion. According to the CPS method, the immunohistochemical assessment of PD-L1 in tumor cells and immune cells is associated with lower pathological T and N stages, the absence of distant metastases, a lower TNM stage, and the absence of lymphatic, vascular, and perineural invasion.

Based on the obtained results, we determined that there are no significant differences between these scoring systems in relation to BRAF and RAS mutations.

We found significant differences in both systems, which showed higher PD-L1 positivity in males younger than 65 years and with localization on the right side.

There are significantly more PD-L1-positive samples with a defect in one or two or more MMR proteins.

We also determined that a better assessment of PD-L1 expressivity occurs for large resection samples due to the existence of staining heterogeneity (diffuse and focal).

With our model of the immunohistochemical assessment of PD-L1 expression using the TPS and the CPS, we tried to show that it is feasible in colorectal cancer. Our results can serve as a basis for further research.

In conclusion, we have shown that CPS is a more useful method of assessing PD-L1 expression than TPS. Colorectal cancers with a higher percentage of immunohistochemical expression of PD-L1 when using the CPS method could have a disorder in the PD-1/PD-L1 immune checkpoint pathway and, as such, represent a subgroup that may benefit from anti-PD-L1 therapy.

We also showed that the immunohistochemical expression of PD-L1, MMR, and MSI status, as well as the status of TILs, are independent prognostic predictors in patients with CRC, so they should be considered in further research.

It should be emphasized that these results should be interpreted in the context of all other factors, such as intrinsic MMR status, microsatellite instability, the existence of intratumoral immune cells, the tumor burden of gene mutations (clonality, aneuploidy, and classes of mutations), and extrinsic factors, such as the microbiome. The patient would benefit from their application [5,15,24,25,26,27]. Immunotherapy is certainly not harmless and has harmful consequences; still, the need for radical surgical procedures could be significantly reduced, as could the severe toxic effects of neoadjuvant chemotherapy with fluoropyrimidine and oxaliplatin, such as intestinal, urinary, and erectile dysfunction, infertility, and sensory neuropathy.

Neoadjuvant radiotherapy and chemotherapy result in a complete response in a quarter of patients. Still, complications after surgery and, unfortunately, a high rate of incomplete clinical responses encourage us to show more interest in investigating a non-operative treatment that spares organs.

Considering the results from previously published studies and the results in our study, for the selection of candidates, we recommend primarily including patients who are younger than 65 years, those of the male gender, those with tumor localization on the right side, those who have a deficiency of one or more MMR proteins, and those with abundant clusters of tumor-infiltrating lymphocytes and abundant clusters of tumor-associated macrophages. We recommend the evaluation of resection samples for better visibility, as it has been proven that PD-L1 expression can be diffuse and focal, and such evaluation is not possible in small biopsy samples [24].

By introducing immunotherapy as an option in the treatment of CRC in a locally advanced stage, the quality of life of patients could be significantly improved without the harmful consequences of chemotherapy and radiotherapy, as well as the psychosocial consequences of living with a colostomy.

This approach would probably be another step closer to the application of personalized therapy in patients with CRC, which we all strive for in the future.

## Figures and Tables

**Figure 1 diagnostics-14-01007-f001:**
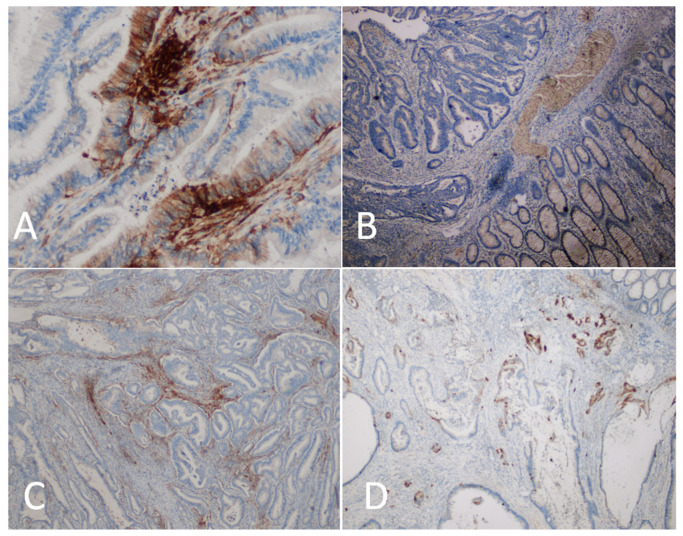
Assessment of the immunohistochemical expression of PD-L1 in colorectal cancer. (**A**)—PD-L1-positive tumor and immune cells (×400). (**B**)—tumor and immune cells negative for PD-L1 (×200). (**C**)—PD-L1-positive immune cells, and negative tumor cells (×200). (**D**)—PD-L1-positive tumor cells, and negative immune cells (×200).

**Figure 2 diagnostics-14-01007-f002:**
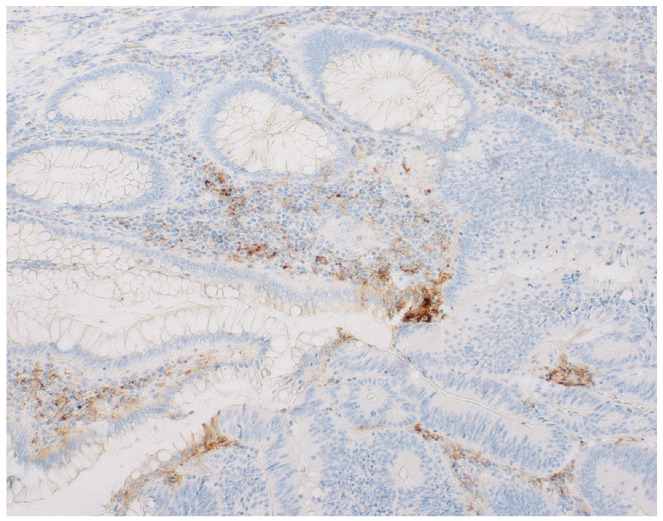
Image of PD-L1 expression at the border of the transition from normal mucosa to tumor tissue.

**Figure 3 diagnostics-14-01007-f003:**
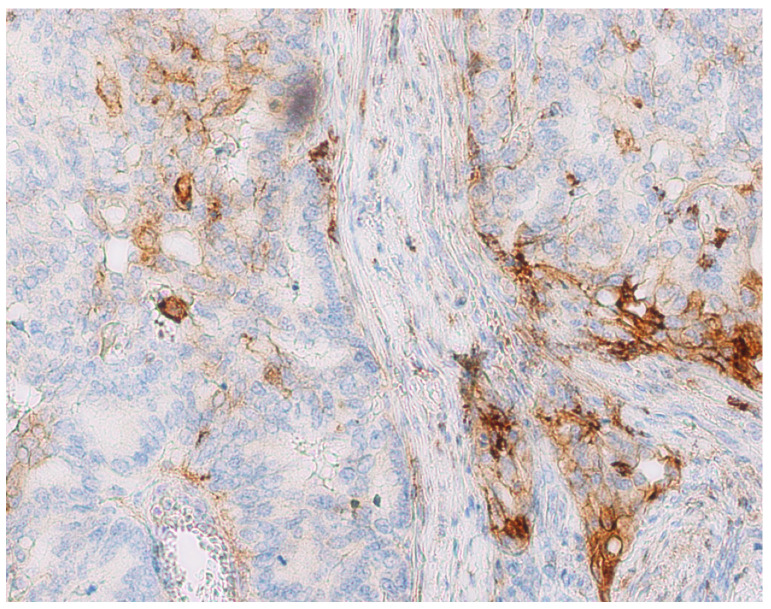
PD-L1 expression in tumor and tumor stroma. TPS: 5–10%, CPS: 10–50% (×400).

**Table 1 diagnostics-14-01007-t001:** Patients according to the TPS and CPS and measures of diagnostic accuracy.

		N (%) TPS			*p*	κ (95% CI) [*p*-Value]
		Negative	Positive	Total		
CPS	Negative	5	0	5 (5)	<0.001	0.07 (0.008 to 0.13) [0.06]
	Positive	55	42	97 (95)		
	Total	60 (59)	42 (41)	102 (100)		
Sensitivity (95% CI)					100% (92–100%)	
Specificity (95% CI)					8.3% (2.8–18.4%)	
Positive predictive value (95% CI)					43.3% (41.4–45.2%)	
Negative predictive value (95% CI)					100% (47.8–100%)	
Positive Likelihood ratio (95% CI)					1.09 (1.01–1.18)	
Negative likelihood ratio (95% CI)					0	
Accuracy (95% CI)					46.1% (36.2–56.2%)	

**Table 2 diagnostics-14-01007-t002:** Distribution of features according to the TPS.

	N (%) of Samples According to the TPS						*p*
	<1%	≥1–5%	≥5–10%	≥10–50%	>50%	Total	
TNM Stage							
I	1 (2)	2 (10)	1 (10)	0	0	4 (4)	0.49
II	16 (27)	5 (24)	4 (40)	2 (22)	1 (50)	28 (27)	
III	37 (62)	11 (52)	4 (40)	4 (44)	1 (50)	57 (56)	
IV	6 (10)	3 (14)	1 (10)	3 (33)	0	13 (13)	
Histology							
Adenocarcinoma	56 (93)	17 (81)	10 (100)	5 (56)	2 (100)	90 (88)	**0.02**
Mucinous	4 (7)	3 (14)	0	4 (44)	0	11 (11)	
Medulllar	0	1 (5)	0	0	0	1 (1)	
Grade							
Well differentiated	1 (2)	0	0	0	0	1 (1)	**<0.001**
Moderately differentiated	54 (90)	16 (76)	9 (90)	2 (22)	2 (100)	83 (81)	
Weak differentiated	5 (8)	5 (24)	1 (10)	7 (78)	0	18 (18)	
Lymph Node Status							
0	38 (63)	18 (86)	7 (70)	2 (22)	2 (100)	67 (66)	**0.005**
1	13 (22)	1 (5)	3 (30)	1 (11)	0	18 (18)	
2	9 (15)	2 (10)	0	6 (67)	0	17 (17)	
Lymphovascular Invasion							
Positive	1 (2)	0	0	0	0	1 (1)	0.37
Negative	18 (31)	5 (24)	3 (30)	6 (67)	0	32 (32)	
Unknown	40 (68)	16 (76)	7 (70)	3 (33)	2 (100)	68 (67)	
Perineural Invasion							
Positive	19 (32)	7 (33)	1 (10)	6 (67)	0	33 (32)	0.10
Negative	41 (68)	14 (67)	9 (90)	3 (33)	2 (100)	69 (68)	
TIL Status							
None	2 (3)	0	0	0	0	2 (2)	0.29
Moderate	45 (75)	16 (76)	4 (40)	6 (67)	1 (50)	72 (71)	
Full	13 (22)	5 (24)	6 (60)	3 (33)	1 (50)	28 (27)	
TAM Status							
Small	53 (88)	18 (86)	8 (80)	8 (89)	2 (100)	89 (87)	0.52
Full	7 (12)	3 (14)	2 (20)	1 (11)	0 (0)	12 (13)	
Metastasis							
None	51 (85)	18 (86)	8 (80)	6 (67)	2 (100)	85 (83)	0.73
Only one organ	8 (13)	3 (14)	2 (20)	3 (33)	0	16 (16)	
Multiple organs	1 (2)	0	0	0	0	1 (1)	

**Table 3 diagnostics-14-01007-t003:** Distribution of features according to the CPS.

	N (%) of Samples According to the CPS						*p*
	<1%	≥1–5%	≥5–10%	≥10–50%	>50%	Total	
TNM Stage							
I	0	1 (11)	1 (3)	1 (4)	1 (4)	4 (4)	0.34
II	3 (60)	0	10 (28)	7 (26)	8 (32)	28 (27)	
III	2 (40)	6 (67)	23 (64)	15 (56)	11 (44)	57 (56)	
IV	0	2 (22)	2 (6)	4 (15)	5 (20)	13 (13)	
Histology							
Adenocarcinoma	5 (100)	7 (78)	33 (92)	26 (96)	19 (76)	90 (88)	0.23
Mucinous	0	2 (22)	3 (8)	1 (4)	5 (20)	11 (11)	
Medulllar	0	0	0	0	1 (4)	1 (1)	
Grade							
Well differentiated	0	1 (11)	0	0	0	1 (1)	0.10
Moderately differentiated	5 (100)	6 (67)	31 (86)	24 (89)	17 (68)	83 (81)	
Weak differentiated	0	2 (22)	5 (14)	3 (11)	8 (32)	18 (18)	
Lymph Node Status							
0	3 (60)	3 (33)	23 (64)	22 (81)	16 (64)	67 (66)	**0.02**
1	2 (40)	2 (22)	10 (28)	1 (4)	3 (12)	18 (18)	
2	0	4 (44)	3 (8)	4 (15)	6 (24)	17 (17)	
Lymphovascular Invasion							
Positive	0	0	0	0	1 (4)	1 (1)	0.12
Negative	0	6 (67)	10 (28)	7 (27)	9 (36)	32 (32)	
Unknown	5 (100)	3 (33)	26 (72)	19 (73)	15 (60)	68 (67)	
Perineural Invasion							
Positive	1 (20)	6 (67)	10 (28)	7 (26)	9 (36)	33 (32)	0.22
Negative	4 (80)	3 (33)	26 (72)	20 (74)	16 (64)	69 (68)	
TIL Status							
None	0	0	2 (6)	0	0	2 (2)	**0.01**
Moderate	4 (80)	9 (100)	29 (81)	18 (67)	12 (48)	72 (71)	
Full	1 (20)	0	5 (14)	9 (33)	13 (52)	28 (27)	
TAM Status							
Small	5 (100)	9 (100)	34 (94)	21 (78)	20 (80)	89 (87)	0.23
Full	0	0	2 (6)	6 (22)	5 (20)	13 (13)	
Metastasis							
None	5 (100)	7 (78)	33 (92)	21 (78)	19 (76)	85 (83)	0.41
Only one organ	0	2 (22)	3 (8)	5 (19)	6 (24)	16 (16)	
Multiple organs	0	0	0	1 (4)	0 (0)	1 (1)	

## Data Availability

Data are unavailable due to privacy or ethical restrictions.

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
