# Peer review of "Evaluation of PD-L1 Expression in Colorectal Carcinomas by Comparing Scoring Methods and Their Significance in Relation to Clinicopathologic Parameters"

_diagnostics, 2024, doi:10.3390/diagnostics14101007_

Round 1
Reviewer 1 Report
Comments and Suggestions for Authors
The manuscript presents the results on the role of PD-L1 in colorectal cancer. It’s an interesting study, however there are several major issues:
- All Figures, including Fig 1, 2 and 3 and their descriptions should be moved to Results sections.
- Methodology sections should be divided into sections for methods used
- Sample characteristics should be moved to Methodology section
- All Correlations should be presented in Figures
Reviewer 2 Report
Comments and Suggestions for Authors
1. What factors might contribute to the discrepancy in PD-L1 positivity rates between the TPS and CPS methods, particularly with a notably higher proportion of negative cases in TPS (58.8%) compared to CPS (4.9%)?
2. Why do samples with 5 to 10% positive cells show the highest percentage of positivity when assessed using the CPS method?
3. What factors could contribute to the substantial discrepancy in positive findings between TPS and CPS methods for PD-L1 expression, with significantly more patients identified as positive using CPS (95%) compared to TPS (41%)
4. What does the low kappa value (κ = 0.07) suggest about the level of agreement between these two assessment methods?
5. What clinical or biological factors might contribute to the significant differences observed between TPS and CPS scores in PD-L1 expression, particularly in relation to gender, age, and tumor localization?
6. How might these findings influence the selection of PD-L1 assessment methods for personalized treatment strategies in different patient subgroups.
7. What underlying mechanisms could explain the observed association between TPS levels and MMR protein deficiency, particularly the higher prevalence of two or more protein deficiencies in patients with TPS values ranging from 1-5% compared to those with TPS < 1% or higher values (10-50%)?
8. What factors might contribute to the lack of significant association between CPS and MMR status in this context
9. How might the observed differences in TPS and CPS percentages between patients with RAS or BRAF mutations versus those without these mutations at various tumor sites inform our understanding of tumor biology and the immune microenvironment in colorectal cancer.
Round 2
Reviewer 1 Report
Comments and Suggestions for Authors
Authors addressed all comments